# *Corynebacterium pseudotuberculosis* Infections in Alpacas (*Vicugna pacos*)

**DOI:** 10.3390/ani12131612

**Published:** 2022-06-22

**Authors:** Reinhard Sting, Claudia Geiger, Wolfram Rietschel, Birgit Blazey, Ingo Schwabe, Jörg Rau, Lisa Schneider-Bühl

**Affiliations:** 1Chemical and Veterinary Analysis Agency Stuttgart (CVUAS), Schaflandstr. 3/3, 70736 Fellbach, Germany; claudia.geiger@cvuas.bwl.de (C.G.); birgit.blazey@cvuas.bwl.de (B.B.); ingo.schwabe@cvuas.bwl.de (I.S.); joerg.rau@cvuas.bwl.de (J.R.); lisa.schneider-buehl@cvuas.bwl.de (L.S.-B.); 2Consiliary Laboratory for Corynebacterium Pseudotuberculosis (DVG), Schaflandstr. 3/3, 70736 Fellbach, Germany; 3Pferdeklinik in Kirchheim, Nürtinger Str. 200, 73230 Kirchheim unter Teck, Germany; rietschelwolf@kabelbw.de

**Keywords:** alpaca, camelid, *Corynebacterium pseudotuberculosis*, caseous lymphadenitis, ELISA, immunoblot, post-mortem examination, antibiotic susceptibility testing, MALDI-TOF MS, FT-IR

## Abstract

**Simple Summary:**

Alpacas have a quite recent history in Europe. Only in the 1990’s were a considerable number of alpacas imported into the UK and then into continental Europe, where the number of animals increased significantly. This development has necessitated a more thorough knowledge of infectious diseases in alpaca husbandry. One of the most serious infectious diseases in alpacas is caseous lymphadenitis (CLA), which is caused by *Corynebacterium* (*C*.) *pseudotuberculosis*. As shown by the post-mortem examinations in this study, CLA is characterized by the formation of multiple, visible abscesses, particularly affecting the lungs and liver as the pathogen spreads throughout the body. While post-mortem examinations and isolation of the pathogen are pivotal for a proven diagnosis, serological examinations are the basis of epidemiological and monitoring studies. We examined 232 alpacas living in three independent herds for this current study. *C. pseudotuberculosis* was able to be isolated in several alpacas. All of the alpacas were tested serologically using a commercially available ELISA, an in-lab ELISA, and, in select animals, immunoblot. The immunoblot showed the highest sensitivity in the early phase of the infection in alpacas proven to be infected with *C. pseudotuberculosis*. Testing for antimicrobial susceptibility based on minimal inhibitory concentration using the broth microdilution method revealed uniform susceptibility to aminopenicillins, cephalosporines, macrolides, enrofloxacin, florfenicol, tetracycline, sulfonamid/trimethoprime, tiamulin, gentamicin, neomycin, spectinomycin, and vancomycin, but resistance to colistin, nitrofurantoin, and oxacillin. The present study shows that *C. pseudotuberculosis* poses an imminent health risk for alpaca husbandry. Laboratory diagnostics based on post-mortem, bacteriological and serological examinations are valuable tools for implementing specific measures as a basis for tackling this infectious disease in alpaca herds.

**Abstract:**

Alpacas are the major camelid species in Europe held for hobbies, animal-aided therapy, and commercial reasons. As a result, health-related issues associated with alpacas are of growing significance. This especially holds true for one of the most serious infectious diseases, caseous lymphadenitis, which is caused by the bacterial pathogen *Corynebacterium* (*C*.) *pseudotuberculosis*. Our study focuses on post-mortem examinations, the laboratory diagnostic tool ELISA, and the immunoblot technique for the detection of specific antibodies against *C. pseudotuberculosis* and detection of the causative pathogen in alpaca herds. We examined a total of 232 alpacas living in three herds. Four of these alpacas were submitted for post-mortem examination, revealing abscesses, apostematous and fibrinous inflammation in inner organs, pleura, and peritoneum. Serological investigation using a commercial ELISA based on phospholipase D (PLD) as antigen and an in-lab ELISA based on whole cell antigens (WCA) revealed an overall seroprevalence of 56% and 61.2%, respectively. A total of 247 alpaca sera originating from 232 animals were tested comparatively using the in-lab and the commercial ELISA and showed a substantial degree of agreement, of 89.5% (Cohen’s kappa coefficient of 0.784), for both tests. Further comparative serological studies using the two ELISAs and the immunoblot technique were carried out on selected sera originating from 12 breeding stallions and six breeding mares for which epidemiological data and partial *C. pseudotuberculosis* isolates were available. The results showed the immunoblot to have a sensitivity that was superior to both ELISAs. In this context, it should be emphasized that evaluation of these investigations and the epidemiological data suggest an incubation period of one to two months. Antibiotic susceptibility testing of 13 *C. pseudotuberculosis* isolates based on the determination of minimal inhibitory concentrations using the broth microdilution method revealed uniform susceptibility to aminopenicillins, cephalosporines, macrolides, enrofloxacin, florfenicol, tetracycline, sulfonamid/trimethoprime, tiamulin, gentamicin, neomycin, spectinomycin, and vancomycin, but resistance to colistin, nitrofurantoin, and oxacillin.

## 1. Introduction

Alpacas have been gaining importance in livestock and companion animal husbandry in Europe since 2000 [1,2]. As a result, the increasing number of alpaca herds has led to more frequent contact between animals originating from different herds due to breeding, restocking, and importing of animals from different countries. Without the establishment of powerful biosecurity management systems, the transmission of infectious diseases between herds will have an increasingly deleterious effect on animal health and welfare. This holds especially true for *Corynebacterium* (*C*.) *pseudotuberculosis*, the pathogenic agent of the chronic and debilitating disease caseous lymphadenitis (CLA). CLA is one of the most important infectious bacterial diseases in camelids. It is already present and gaining ground in camelids in Europe, resulting in an increase in herds affected by this insidious infectious disease. Thus, CLA in livestock can lead to serious economic losses and impairment of animal health and welfare [3,4].

*C. pseudotuberculosis* causes abscesses of external and internal lymph nodes and inner organs of sheep and goats as well as those of Old World and New World camelids [4]. The final stage of the disease is characterized by general wasting and emaciation of the infected animals [5,6].

Infections are mainly caused by the transmission of the pathogen through wounds, inhalation, or ingestion [4]. Diagnosis of CLA in camelids is difficult due to an uncertain incubation period and infestation of inner organs [4,5,7]. Tools for diagnosing CLA include clinical and post-mortem examinations, direct detection of the pathogen by cultivation or polymerase chain reaction (PCR), and detection of specific antibodies by serological testing based on the enzyme-linked immunosorbent assay (ELISA) technique [5,8,9,10,11]. While the immunoblot technique has been used for goat and sheep sera in a comparative study using ELISA and immunoblot [12], there is no experience with camelids. These diagnostic tools are essential and complementary components for the control of CLA. They focus on determining the CLA status of a herd, removing infected animals from a herd, pre-screening for the detection of infected animals prior to purchase of animals from herds of unknown CLA status, and monitoring animals in quarantine. Thus, powerful biosecurity measures, monitoring, control, and prevention are all paramount for combating this infectious disease because CLA is currently considered incurable, and vaccination does not yet ensure sustainable success [3,4].

In 2020 we noted an abrupt increase in samples from alpacas for laboratory diagnostic of *C. pseudotuberculosis*, a high proportion of which tested positive. Hence, the current situation raises the need for awareness of CLA in alpacas as an increasingly important disease in this camelid species.

The aim of the present paper is thus to provide data on *C. pseudotuberculosis* infections in alpaca herds, focusing on pathological-anatomical findings, clinical signs, the incubation period, and laboratory diagnostics as a basis for implementing control and prevention measures for CLA.

## 2. Materials and Methods

### 2.1. Alpaca Herds and Samples

In the current study, three alpaca herds (herds A, B, and C) located in Germany, with a total of 232 animals, were investigated. Herd A consisted of about 200, herd B of about 30, and herd C of about 20 alpacas. Samples from all animals in the three herds older than six months were taken. The alpacas were being kept for commercial farming. Due to contact between animals from herds A and B, related data were requested, and diagnostic testing was pursued as much as possible.

In these herds, nine *C. pseudotuberculosis* isolates were able to be cultured from living animals, and four isolates were taken from animals that had been subjected to post-mortem examinations. For serological examinations, a total 247 sera were taken from 232 alpacas.

### 2.2. Serological Examinations

For serological examinations, both the commercially available ELISA ID Screen^®^ *Corynebacterium pseudotuberculosis* Indirect (Innovative Diagnostics (IDvet), Grabels, France) based on a phospholipase D (PLD) antigen (PLD ELISA) and an in-lab ELISA based on whole-cell antigens (WCA ELISA) obtained from the ovine *C. pseudotuberculosis* isolate CVUAS 3365 was used and carried out as previously described [6,13]. The cut-off values (S/P% = OD sample—OD negative control/OD positive control—OD negative control) for the PLD ELISA were set at ≤40% negative, 40–50% questionable, and ≥50% positive. For the WCA ELISA, the cut-off values were set at ≤5 MONA (multiple of normal activity) negative, 5–10 MONA questionable, and ≥10 positive [13].

Sera from alpacas suffering from CLA verified by culture and sera originating from animals that had had contact with infected animals were additionally subjected to an immunoblot based on the same whole-cell antigen that had been used for the in-lab WCA ELISA. For the immunoblot, the proteins were separated by sodium dodecyl sulfate polyacrylamide gel electrophoresis (SDS-PAGE) in accordance with standard procedures [14]. In brief, each slot was loaded with 8 µg protein [12] as determined by a Bradford protein assay (DC Protein Assay, Bio-Rad Laboratories, Munich, Germany), and the proteins were separated in a 12% SDS gel. The separated proteins were subsequently transferred onto nitrocellulose membranes (Supported Nitrocellulose Membrane 0.22 µm, Bio-Rad Laboratories, Munich, Germany) using a semi-dry electrophoretic transfer system (Trans Blot, Bio-Rad Laboratories, Munich, Germany), as described by Hoelzle et al. [12]. After immunoblotting, unspecific binding sites on the membrane were blocked with ROTIBlock, diluted 1:10 (Carl Roth, Karlsruhe, Germany). The proteins were then made visible via the binding of antibodies (serum dilution 1:20) and immunostaining using peroxidase-labeled protein G (diluted 1:5000; Calbiochem, Merck, Darmstadt, Germany), thus developing a color reaction with H_2_O_2_ and 4-chloro-1-naphthol. For this, the membrane was cut into strips and incubated using a slot blot device.

Serum from an alpaca suffering from a *C. pseudotuberculosis* abscess was used as a positive control, and serum from an alpaca originating from a herd without any clinical signs of CLA and with negative results from an ELISA served as a negative control.

In total, five stallions (S8, S9, S10, S11, S12) and one mare (M6) were tested several times (paired sera), and seven stallions (S1, S2, S3, S4, S5, S6, S7) and five mares (M1, M2, M3, M4, M5) were tested only once.

### 2.3. Bacteriological Examinations

For the bacteriological examinations, swabs from abscess and organ samples were examined in accordance with standard bacteriological procedures using sheep blood agar (Columbia agar with 5% sheep blood, BD, Heidelberg, Germany) and MacConkey agar (MacConkey II agar, BD, Heidelberg, Germany).

Identification of corynebacterial cultures at the species level was carried out in the first step by MALDI-TOF MS analysis (matrix-assisted laser desorption/ionization mass spectroscopy; Bruker Daltonics, Bremen, Germany), employing the MALDI-Biotyper System (Biotyper software Version 3.1, Bruker Daltonics). This was expanded by an in-house database as previously described [15] and updated with data obtained from *C. silvaticum*, *C. ulcerans*, and *C. pseudotuberculosis* [16]. Further identification was carried out by FT-IR analyses (Bruker Optics, Ettlingen, Germany). Characterization of the isolates was made possible by creating a dendrogram using FT-IR spectra [6].

The presence of the PLD gene was tested by real-time PCR [11], whose production was proven based on synergistic and antagonistic hemolytic interactions produced by the *C. pseudotuberculosis* isolates in the presence of *Rhodococcus* (*R.*) *equi* and *Staphylococcus* (*S.*) *aureus*, respectively. To demonstrate PLD activity *R. equi* (ATCC 33701) and *S. aureus* (ATCC 25923) were inoculated in the center of a sheep blood agar plate (Columbia agar with 5% sheep blood, BD, Heidelberg, Germany), and the *C. pseudotuberculosis* isolates to be tested radially streaked out to the center [17]. In addition, the *C. pseudotuberculosis* isolates were tested for reduction of nitrate, following standard procedure [18].

Antibiotic susceptibility testing of the *C. pseudotuberculosis* isolates was based on determination of minimum inhibitory concentrations (MIC) using the microdilution method. Commercially available antibiotic microdilution plates were used for this purpose in accordance with the manufacturer’s instructions (Micronaut-S Veterinary Large Animal and Microanut-S Lifestock-Equines (GP), MERLIN, Bornheim-Hersel, Germany).

### 2.4. Post-Mortem Examinations

Two animals from herd A and one animal, each from herds B and C, suffered from poor health and visible swellings in the skin or enlargement of the superficial lymph nodes. Two of the alpacas died, and the other two animals were eventually euthanized for reasons of animal welfare. The four alpacas were subjected to post-mortem examinations, and samples were taken for bacteriological and, in two cases, blood samples for serological investigations.

### 2.5. Statistical Analyses

The Cohen’s kappa coefficient value was calculated to assess agreement of the two ELISAs using the free accessible internet program WinEpi (http://www.winepi.net/menu1.php) (accessed on 2 March 2022) [19].

## 3. Results

### 3.1. Herd Study

Animals from all three alpaca herds exhibited clinical signs of CLA, i.e., skin lumps or swelling of superficial lymph nodes (Figure 1). CLA was proven by isolation of the pathogen *C. pseudotuberculosis* from superficial abscesses and abscesses in inner organs during post-mortem examinations (Table 1 and Table 2, Figure 1 and Figure 2). The animals showed positive results in serological testing using the commercial PLD ELISA and the in-lab WCA ELISA, as well as in immunoblot analyses. The animals had been selected for testing by ELISA and immunoblot due to available epidemiological data and/or cultivation of *C. pseudotuberculosis* isolates from abscesses (Table 1, Figure 3).

Of particular interest are alpaca herds A and B because of intermingling among alpacas from both herds. Epidemiological inquiries revealed that stallions S1 and S2 had been newly introduced into herd A in April 2020 without clinical signs of CLA but developed abscesses for the first time three to four months later in July and August 2020, respectively. Both stallions revealed positive serological reactions in September 2020. Stallion S1 attracted attention due to the repeated formation of multiple abscesses three months later in December 2020, and had to be euthanized for reasons of animal welfare on 14 December 2020 (Table 2). Stallion S2 had a negative result from the PLD ELISA but an explicitly positive reaction to the in-lab WCA ELISA, verified by immunoblot on 21 September 2020 (Figure 2). This stallion had mated with mare M7 on 31 August 2020, who subsequently developed abscesses and died about six weeks later on 10 October 2020, due to multiple abscesses caused by *C. pseudotuberculosis* (Table 2). Another mare, M1, developed abscesses in September 2020, verified by isolation of *C. pseudotuberculosis* and serological testing (Table 1).

On 21 August 2020, five alpaca stallions (S3, S4, S5, S6, and S7) originating from herd A were integrated into herd B. All five stallions were tested with identical strong seropositive results in ELISA and an immunoblot two months later on 20 October 2020 (Table 1, Figure 2). Stallion S3 developed an abscess on 20 October 2020 that was caused by *C. pseudotuberculosis*, verified by the cultivation of the pathogen, and stallion S4 was in poor health. Another stallion (S8) was born on 19 May 2018 into herd A and introduced into herd B on 2 November 2018. Due to proven cases of CLA in herds A and B, this stallion was tested on 20 October 2020, revealing a negative serological result. However, seven weeks later, on 9 December 2020, an abscess on the upper jaw below the eye caused by *C. pseudotuberculosis* was noticed in this stallion (Figure 1), who tested clearly seropositive on that date as well as on 15 April and 21 September 2021. *C. pseudotuberculosis* was isolated from an abscess on a further stallion (S9) in herd B on 20 October 2020, and positive serological results were obtained from the in-lab ELISA and immunoblot. The epidemiological inquiries showed that this stallion had been shorn together with three other stallions (S10, S11, S12) in May 2020; these three stallions were therefore also tested by ELISA and immunoblot on 20 October 2020. Stallion S10 tested clearly positive, and stallions S11 and S12 remained seronegative in ELISA but revealed weak positive reactions in the immunoblot (Table 1, Figure 2).

Sera from five stallions (S8, S9, S10, S11, and S12) were retested about half a year later, and for two of them, who had remained in herd B (S8 and S9), again after nearly one year. Stallions S8, S9, and S10 showed extremely positive results with the ELISA and immunoblot, while stallions S11 and S12 remained seronegative in both ELISAs but showed weak reactions in the immunoblot throughout this period (Table 1, Figure 2).

In herd C, *C. pseudotuberculosis* was able to be isolated from one alpaca mare (M2) that had died and from four living alpaca mares (M3, M4, M5, and M6). Blood samples were available from these mares for serological testing and revealed positive results from the ELISA and/or immunoblot. For details, see Table 1 and Figure 2. Alpaca mare M6 had been imported from Peru and introduced into herd C on 12 September 2021. Four weeks later, on 12 October 2021, this mare developed an abscess that proved to have been caused by *C. pseudotuberculosis* (CVUAS 33314) and reacted seropositive to ELISA and immunoblot, except for the first and second testing in the PLD ELISA (Table 1, Figure 2).

The period of time between contact with infected alpacas and the development of abscesses (M6, M7) or serological reactions (S8) was found to be about one to two months.

### 3.2. Post-Mortem Examinations

Four alpacas in farms A, B, and C were suffering from poor health and had attracted attention due to visible swellings in the skin or enlargement of the superficial lymph nodes. These clinical signs lead to suspicion of CLA. Two animals were euthanized because of poor health and body condition (S1, M8), and two animals died (M2, M7) and were subjected to post-mortem examinations. These post-mortem examinations revealed abscesses in the subcutis, superficial lymph nodes, and inner organs, summarized in Table 2. Abscesses in the liver and spleen detected during the post-mortem examination of stallion S1 are shown in Figure 2 to illustrate the visual appearance of abscesses in inner organs during post-mortem examinations. The described inflammatory conditions were confirmed by histopathological examinations.

### 3.3. Bacteriological Examinations

*C. pseudotuberculosis* isolates were cultivated from abscess material taken from a total of 13 alpacas originating from the three herds A, B, and C. Nine isolates were obtained from living animals displaying visible formation of abscesses in the skin or lymph nodes, and four *C. pseudotuberculosis* isolates came from internal abscesses obtained during post-mortem examinations (Table 1 and Table 2).

All isolates were identified to be *C. pseudotuberculosis* and were clearly distinguishable from the most closely related *Corynebacterium* species of the *C. diphtheriae* group (*C. diphtheria*, *C. belfantii*, *C. rouxii*, *C. silvaticum*, *C. ulcerans*) using MALDI-TOF MS and FT-IR. The comparison of IR-spectra of the *C. pseudotuberculosis* isolates revealed a clear separation of *C. pseudotuberculosis* from the other corynebacterial species of the *C. diphtheriae* group. Furthermore, in the FT-IR dendrogram, four separate clusters can be differentiated: a cluster including *C. pseudotuberculosis* isolates originating from sheep and goats, a cluster including isolates from alpacas and dromedaries, a further cluster comprising isolates from alpacas, llama, dromedaries, sheep and a horse, and another cluster containing an isolate from a goat and a llama, respectively (Figure 4).

The isolates proved to belong to the biotype ovis due to the lack of reduction of nitrate. All *C. pseudotuberculosis* isolates carried the PLD gene, verified by the PLD gene-specific real-time PCR, and showed PLD activity in vitro. After simultaneous growth of these bacteria on sheep blood agar, all strains thus exhibited synergistic or antagonistic hemolytic interactions between *C. pseudotuberculosis* isolates and *R. equi* or *S. aureus*, respectively.

The results from determining minimum inhibitory concentrations (MIC) using the microdilution method carried out by microtiter plates are shown in Table 3. Except for colistin, nitrofurantoin, oxacillin, and penicillin, all other tested antibiotics proved effective against *C. pseudotuberculosis* in vitro.

### 3.4. Comparative Serological Examinations

The cases of CLA in alpacas from farms A and B, which had come in contact with each other, as well as the cases from herd C, prompted us to carry out further serological examinations using the PLD and WCA ELISA. In total, 247 serum samples taken from 232 animals were tested to check the *C. pseudotuberculosis* status in the herds. The detailed results are shown in Table 4. These sera were also used for comparative purposes of the PLD and the WCA ELISA. The results of this comparative serological study are shown in Table 5. A measure of agreement between the tests was calculated using the Cohen’s kappa coefficient. The PLD and WCA ELISA yielded a Cohen’s kappa coefficient of 0.784 (observed agreement 89.5%, confidence interval 95%), which, according to Landis and Koch [20], should be interpreted as a substantial agreement. Differing results between the PLD ELISA and WCA ELISA are mainly attributable to sera that tested positive in the WCA ELISA but negative in the PLD ELISA (*n* = 21). In contrast, five serum samples revealed positive results in the PLD ELISA but negative reactions in the WCA ELISA (Table 5).

PLD ELISA (enzyme-linked immunosorbent assay) = ELISA ID Screen^®^ *Corynebacterium pseudotuberculosis* Indirect (Innovative Diagnostics (Idvet), Grabels, France) based on a phospholipase D (PLD) antigen; WCA ELISA = in-lab ELISA based on whole-cell antigens.

## 4. Discussion

*C. pseudotuberculosis* infections have been described in Peruvian alpacas as early as 1985 [21] and later in Germany [22]. Nevertheless, it was only in 2020 that we abruptly received more samples for testing of *C. pseudotuberculosis* and carcasses for post-mortem examinations, reflecting an increased occurrence of CLA in alpacas and greater awareness among owners and veterinarians.

### 4.1. Post-Mortem Examinations

Based on clinical and post-mortem examinations, our study confirms that *C. pseudotuberculosis* can cause chronic, severe, debilitating, and, in the long term, lethal infections in alpacas due to the development of abscesses in inner organs. All four animals that had been submitted for post-mortem examinations had shown clinical signs of CLA, represented by visible external abscesses in the skin and in superficial lymph nodes. These alpacas also had abscesses in inner organs, predominantly in the lungs, but also in the liver, spleen, or kidney. Moreover, these animals also showed fibrinous pleuritic or peritonitis and apostematous changes of the heart.

In summary, *C. pseudotuberculosis* was able to be isolated from central organs in high loads from all dissected carcasses. In accordance with our findings, other researchers also reported abscesses in the superficial, mesenterial, and internal renal lymph nodes and detected internal abscesses in the central organs, including the lungs, liver, and kidney of all animals that had been examined post-mortem [5,7,8,9].

Pathohistological examinations revealed purulent infections. In contrast to sheep and goats, pathohistological examinations of affected tissues of camelids displayed pyogranulomatous lesions in various forms ranging from purulent to pronounced granulomatous inflammatory processes [5,6].

### 4.2. Bacteriological Examinations

Cultivation of the pathogen *C. pseudotuberculosis* is considered the gold standard for confirmation of CLA. This makes correct identification of the bacteria all the more important. Precise differentiation of corynebacterial species belonging to the *C. diphtheriae* group and reliable differentiation and assignment of other bacteria causing the formation of abscesses and also belonging to the order of Actinomycetales are relevant with regard to treatment and measures for preventing the spread of infections in the herd [23]. MALDI-TOF MS and FT-IR analyses have proven to be reliable and rapid techniques for accurate bacterial identification at the species level [24]. With help from these spectroscopic techniques, the corynebacterial isolates cultivated from the abscesses and organs were unequivocally identified as *C. pseudotuberculosis*. Furthermore, the 13 alpaca *C. pseudotuberculosis* isolates were able to be allocated to four clusters using FT-IR, including isolates originating from llamas, dromedaries, goats, sheep, and a horse, respectively. The majority of the *C. pseudotuberculosis* isolates originating from alpacas cluster together with *C. pseudotuberculosis* isolates from dromedaries, except for two isolates which create a cluster including *C. pseudotuberculosis* isolates from a llama, a dromedary, sheep, and a horse. However, details on relationships between alpaca isolates and those originating from camelids, goats, and sheep are beyond the scope of this study and should be investigated in-depth in a further study.

Testing on antibiotic susceptibility yielded uniform results for all 13 *C. pseudotuberculosis* strains. Aminopenicillins, cephalosporines, macrolides, rifampin, sulfonamide/trimethoprim, tetracyclines, aminoglycosides, tiamulin, and vancomycin proved effective against the vast majority of the *C. pseudotuberculosis* isolates tested by the broth microdilution method. However, colistin, oxacillin, and penicillin G yielded only poor efficacy against all *C. pseudotuberculosis* strains. These results are in close agreement with findings presented by Schulthess [25] in a comprehensive study on antibiotic susceptibility of *C. pseudotuberculosis*. This researcher used breakpoints for the interpretation of the MIC values based on data provided for *Staphylococcus* (*S.*) *aureus* by the Clinical and Laboratory Standards Institute (CLSI). Further studies on the determination of MIC values for aminopenicillins and penicillin G, cephalosporines, macrolides, gentamicin, rifampin, sulfonamide/trimethoprim, and tetracyclines obtained by the broth microdilution method revealed MIC values that deviated from our study and from each other by only one dilution level [26,27,28,29]. However, it must be considered that antibiotics proving effective in vitro may appear ineffective in vivo. Reasons for this include, among others, abscess walls that hamper tissue penetration of antibiotics or barriers analogous to biofilms that multiply effective antibiotic concentrations [30,31]. Nevertheless, an antimicrobial therapy using rifamycin (rifampicin, rifampin) combined with oxytetracycline proved to be effective for the treatment of CLA in sheep [31].

### 4.3. Serological Examinations

In addition to the cultivation of *C. pseudotuberculosis*, CLA in alpacas was verifiable through positive serological reactions in the commercial PLD ELISA, the in-lab WCA ELISA, and immunoblot. This holds true for all alpacas with caseous lymphadenitis verified by culture in our study. This is especially relevant for the testing of herds, both for detecting individual infected animals and for determining the infectious status of the entire herd [3,4].

Tests for detecting antibodies against PLD or whole cell antigen preparation produced by disruption of bacterial cells have already been proven successful for serological testing of alpacas [3,8,9,10]. Braga et al. [8] detected serum antibody titers in ELISA on day 16 after experimental infection by using whole-cell antigens obtained by sonically disrupted *C. pseudotuberculosis* bacteria. Braga et al. [10] also reported on a robust antibody response against PLD in infected alpacas.

A comparison of the two ELISAs used in our study revealed more serologically positive alpacas with the WCA ELISA than with the PLD ELISA. Comparable investigations on dromedaries using ELISA based on PLD and WCA have been presented by Borham et al. [32]. Using a WCA ELISA resulted in 3% more serologically positive animals and an additional 27% seropositive animals without symptoms of CLA. Wernery and Kinne [4] reported a proportion as high as 40% of serologically positively tested New World camels exhibiting no clinical signs; they attributed this finding to the contact the animals had had previously with the pathogen or with carriers having no clinical signs. In our study, we also determined high prevalence rates of 20–70% per herd, which far exceeds the number of alpacas with clinical signs. Therefore, serological screening using ELISA is considered a helpful tool for the detection of alpacas that have had contact with the *C. pseudotuberculosis* pathogen prior to the detection of clinical signs [3,4,8].

A closer look at the results of a comparison between the commercial PLD ELISA and the WCA ELISA showed strong agreement between the two, as interpreted by Landis and Koch [20]. Moreover, the application of the immunoblot technique for serological testing of alpaca sera was carried out in this study for the first time.

The immunoblotting technique has proven here to be a powerful tool for verifying positive results in ELISA and, in particular, for clarifying weak, questionable, and differing results in both ELISAs. Immunoblotting allows the allocation of reactive bands to immuno-dominant proteins, which creates recognition patterns [12,33]. For example, one mare (M3) in our study showed a weak positive serological reaction only in the immunoblot, and another mare (M4) had a positive result in the PLD-ELISA, verified by a weak reaction in the immunoblot, even though *C. pseudotuberculosis* was able to be cultivated in both cases during bacteriological examinations. The sensitivity of the immunoblot thus seems to be higher than both the PLD and the WCA ELISA in our study. However, it has to be considered that in ELISA, seronegative sera showed in two cases weak positive results in immunoblot (stallion S11 and S12). This may indicate contact of these alpacas to the pathogen without clinical significance.

Seropositive sera of the alpacas showed nearly identical immunoblot patterns generated by immunoreactive antigens, with molecular weights ranging from about 20 to 65 kDa. Hoelzle et al. [12] and Paule et al. [33] also detected immunoreactive proteins within this range of molecular weights. However, Hoelzle et al. [12] presented variability in antigen recognition patterns using goat and ovine sera. Further validation studies are necessary to verify our results and to allow for the application of the immunoblot technique in routine laboratory diagnostics for alpaca sera.

In conclusion, serological testing of alpaca herds using ELISA and immunoblots for verification purposes should be implemented as a key element in the control of CLA in alpaca herds. This approach has also been recommended by Anderson et al. [3] and Wernery and Kinne [4].

### 4.4. Source of Infection, Incubation Period

In our case, the introduction of the stallions S1 and S2 into herd A in April 2020 might have been the cause of the *C. pseudotuberculosis* infections that subsequently developed, as both stallions developed abscesses three to four months later. The source of infection in herd B is assumed to have occurred through the purchase of five alpaca stallions (S3, S4, S5, S6, and S7) from herd A in August 2020. In herd B, stallion S9 developed *C. pseudotuberculosis* abscesses two months later, in October 2020. This epidemiological information shows that, without implementing any biosecurity measures, the integration of new animals infected with *C. pseudotuberculosis* into a CLA-free herd can cause CLA after only a few months. This phenomenon has also been observed in other studies. The onset of the disease is reported to be about 40 to 60 days after infection, resulting in the development of superficial abscesses [3] or internal abscesses [8], respectively.

Transmission of the pathogen occurs via inhalation, ingestion, or through wounds of the thin skin of alpacas [3,4]. Moreover, contamination of the environment after the rupturing of abscesses and the high robustness of the pathogen enable its survival in the environment for a month or even longer [3,34]. Transmission via insects should also be considered a relevant factor [11]. The import and purchase of infected animals with the resulting transmission of *C. pseudotuberculosis* via direct alpaca-to-alpaca contact, e.g., during breeding, are considered the most important reasons and risks for the spread of CLA among herds [3,4,5].

An incubation period of six to seven weeks between a possible first contact with infected animals and the onset of disease, as seen in *C. pseudotuberculosis* in abscesses or serological reactions in alpacas, was observed for two alpacas (M7, S8). Anderson et al. [3] also reported on the detection of abscesses in an alpaca that had been brought into a *C. pseudotuberculosis* infected herd three months earlier. However, in our study, the time interval between the introduction of mare M6 into herd C and the formation of abscesses and production of antibodies was only four weeks.

### 4.5. Treatment and Control of CLA

Control of CLA in alpaca herds is of particular importance because the success of antimicrobial treatment is poor, despite sufficient in vitro susceptibility of this pathogen. Nevertheless, vaccination of alpacas using commercial or autogenous vaccines provides variable protection [4,9]. Anderson et al. [3] did not favor vaccination in affected herds, however, due to seroconversion and unknown side effects, but rather recommended isolation of infected animals, monitoring of the herd, and serological survey as a non-invasive technique. This procedure includes key actions according to guidelines created for combating CLA in goats [13].

## 5. Conclusions

CLA in alpacas is a severe, life-threatening disease, the control of which is urgently needed. Measures to prevent *C. pseudotuberculosis* infections should be implemented in accordance with CLA guidelines for goats and sheep. The most important measures are regular clinical and serological monitoring at the herd level and the testing of animals that are intended to be introduced into a herd, supported through appropriate hygiene programs and separation of infected animals.

## Figures and Tables

**Figure 1 animals-12-01612-f001:**
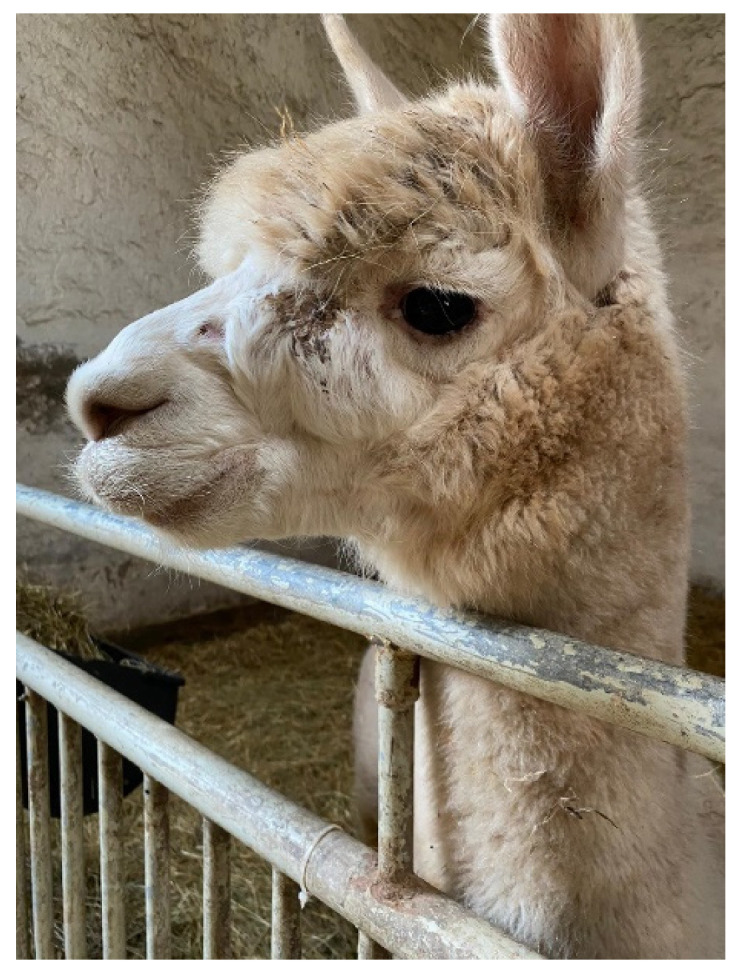
Abscess on the upper jaw below the eye caused by *C. pseudotuberculosis* (stallion S8).

**Figure 2 animals-12-01612-f002:**
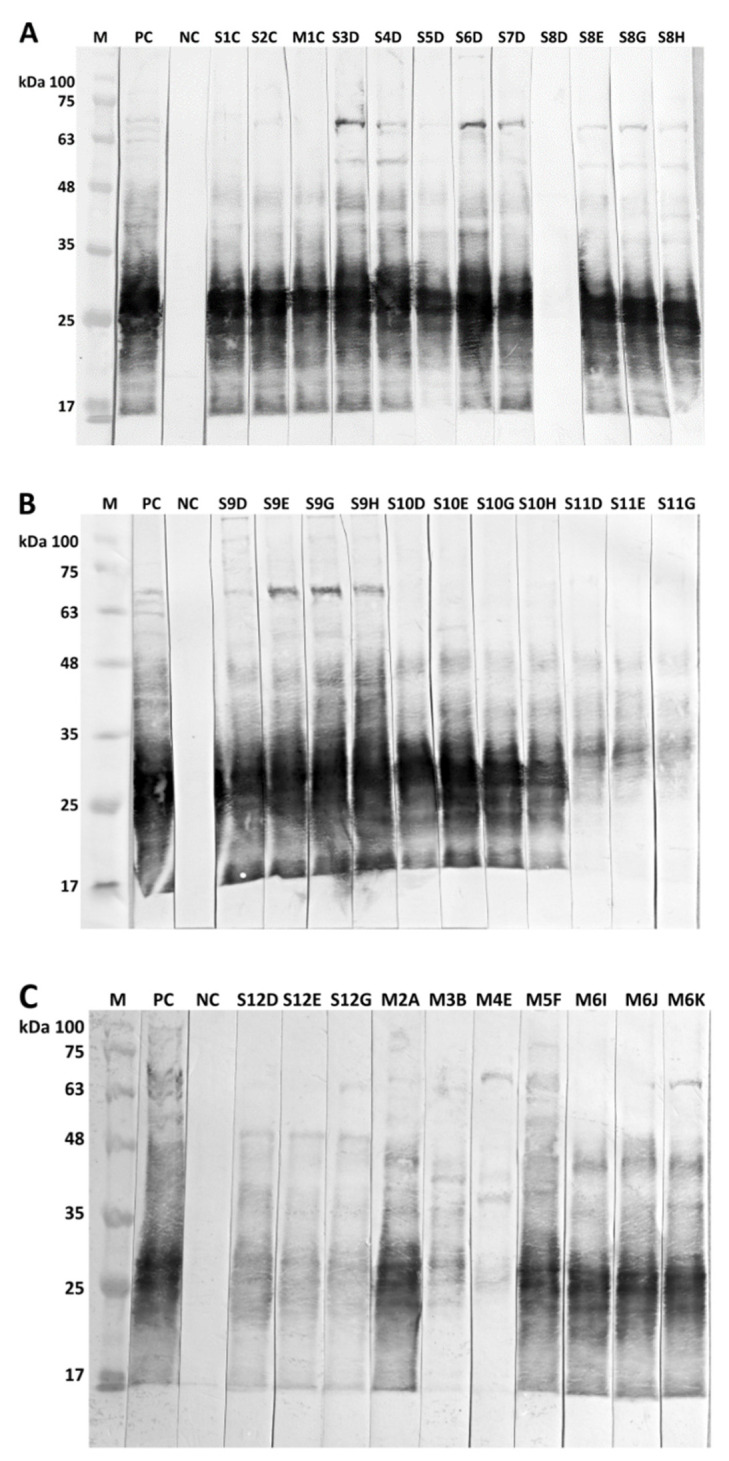
Immunoblot analyses of 33 blood samples originating from 17 alpacas for detection of antibodies against *C. pseudotuberculosis*. See Table 1 for details on the alpacas. Immunoblot (**A**) Stallions S1–S8, mare M1; Immunoblot (**B**) Stallions S9–S11; Immunoblot (**C**) Stallion S12, mares M2–M6; Sampling dates: A: January 2020, B: March 2020, C: September 2020, D: October 2020, E: December 2020, F: February 2021, G: April 2021, H: September 2021, I: October 2021, J: November 2021, K: December 2021; M = marker (kDa = kilo Dalton), PC = positive control, NC = negative control.

**Figure 3 animals-12-01612-f003:**
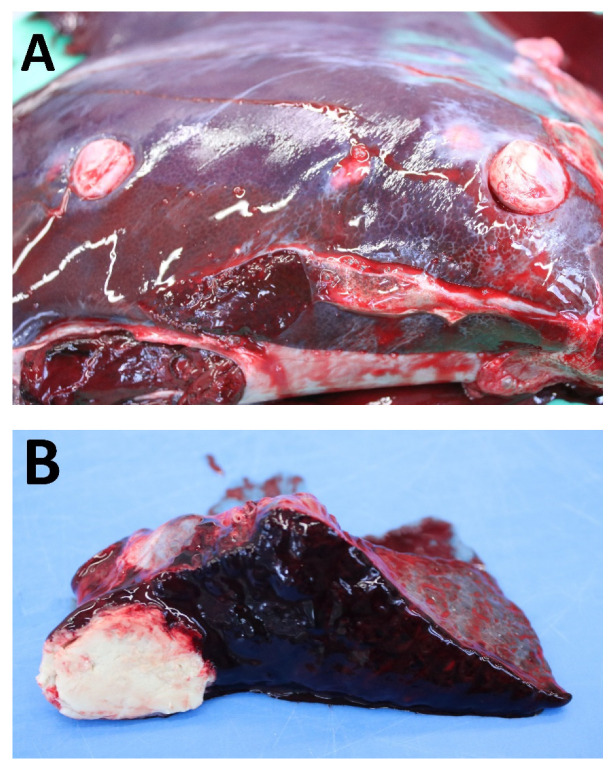
Abscesses in the liver (**A**) and the spleen (**B**) of stallion S1. The abscesses were caused by *C. pseudotuberculosis* and detected during post-mortem examination.

**Figure 4 animals-12-01612-f004:**
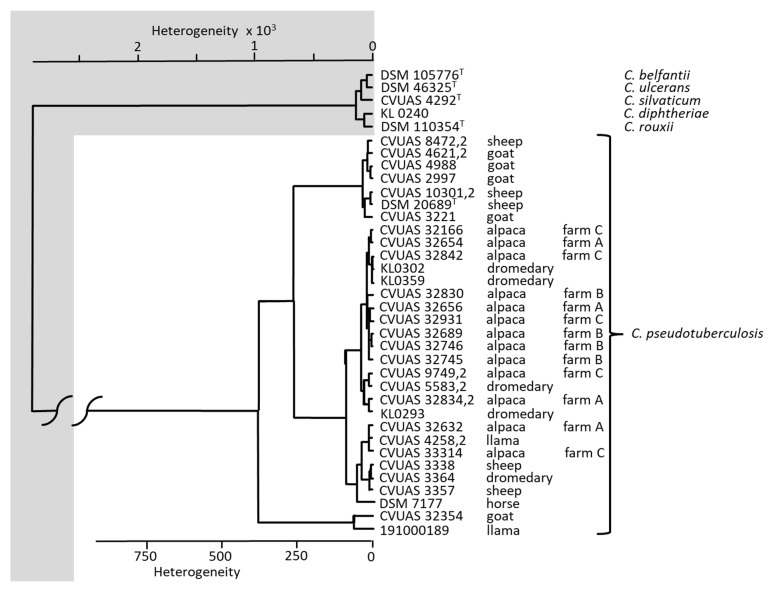
Dendrogram of IR-spectra of *Corynebacterium (C.) pseudotuberculosis* isolates from alpacas in comparison with spectra from dromedaries, llamas, goats and sheep. Isolates of *C. ulcerans*, *C. silvaticum*, *C. rouxii*, *C. diphtheriae* and *C. belfantii* were included for differentiation of the corynebacterial species belonging to the *C. diphtheriae* group.

**Table 1 animals-12-01612-t001:** Results of bacteriological and serological examinations on caseous lymphadenitis in selected alpacas in three herds. The name of the *C. pseudotuberculosis* isolates and the date of isolation are given in brackets in the first column.

Animal	Herd	Date of Sampling for Serological Testing	Result PLD-ELISA	Result WCA-ELISA	Result Immunoblot
Stallion S1 ^1,2^ euthanized on 14 December 2020 (CVUAS 32834.2)	A	21 September 2020	–	+	+
Stallion S2 ^1,3^	A	21 September2020	–	+	+
Mare M1 ^2^ (CVUAS 32654, 22 September 2020)	A	22 September 2020	+	+	+
Stallion S3 ^1,2,4^ (CVUAS 32746, 20 October 2020)	B	20 October2020	+	+	+
Stallion S4 ^4,5^	B	20 October 2020	+	+	+
Stallion S5 ^4,6^	B	20 October 2020	+	+	+
Stallion S6 ^4,6^	B	20 October 2020	+	+	+
Stallion S7 ^4,6^	B	20 October 2020	+	+	+
Stallion S8 ^2^ (CVUAS 32830, 9 December 2020)	B	20 October2020	–	–	–
		9 December 2020	+	+	+
		15 April 2021	+	+	+
		21 September 2021	+	+	+
Stallion S9 ^2,7^ (CVUAS S32745, 20 October 2020)	B	20 October 2020	–	+	+
		9 December 2020	+	+	+
		15 April 2021	+	+	+
		21 September 2021	+	+	+
Stallion S10 ^7^	B	20 October 2020	–	+	+
		9 December 2020	+	+	+
		15 April 2021	+	+	+
		21 September 2021	+	+	+
Stallion S11 ^7^	B	20 October 2020	–	–	+(weak)
		9 December 2020	–	–	+(weak)
		15 April 2021	–	–	+(weak)
Stallion S12 ^7^	B	20 October 2020	–	–	+(weak)
		9 December 2020	–	–	+(weak)
		15 April 2021	–	–	+(weak)
Mare M2 ^2^ died on 16 January 2020 (CVUAS 32166)	C	15 January 2020	+	+	+
Mare M3 ^2^ (CVUAS 9749.2, 6 March 2020)	C	6 March 2020	–	–	+(weak)
Mare M4 ^2^ (CVUAS 32842, 11 December 2020)	C	11 December 2020	+	–	+(weak)
Mare M5 ^2^ (CVUAS 32931, 2 February 2021)	C	8 February 2021	+	+	+
Mare M6 ^2^ (CVUAS 33314, 8 October 2021)	C	8 October 2021	–	+	+
		3 November 2021	–	+	+
		1 December 2021	+	+	+
Mare M7 ^2^ died on 10 October 2020 (CVUAS 32689)	B	/	n.t.	n.t.	n.t.
MareM8 ^2^ euthanized on 24 September 2020 (CVUAS 32656)	A	/	n.t.	n.t.	n.t.
Mare M9 ^2^ (CVUAS 32632, 12 September 2020)	A	/	n.t.	n.t.	n.t.

^1^ Stallion that had been purchased and simultaneously introduced into herd A in April 2020; ^2^ Alpaca with abscesses and isolation of *C. pseudotuberculosis*; ^3^ Alpaca with abscesses but without bacterial examination; ^4^ Stallion which had been introduced from herd A into herd B on 21 August 2020; ^5^ Stallion in poor health condition. ^6^ Animal without clinical signs. ^7^ Stallion that had been shorn together in May 2020. N.t. = not tested serologically, only by bacterial examination.

**Table 2 animals-12-01612-t002:** Superficial abscesses and internal abscesses detected by post-mortem examinations. The names of the *C. pseudotuberculosis* isolates are given in brackets in the first column.

Animal, Herd	Herd	Date	Visible Abscesses in	Absecces in			
				Lungs	Liver	Spleen	Kidney
Stallion S1 ^1^ (CVUAS 32834.2)	A	Euthanized on 14 December 2020	Ln. cervicalis superficialis	−	+	+	−
Mare M2 ^2,3^ (CVUAS 32166)	C	Died on 16 January 2020	Ln. mandibularis, Ln. subiliacus	+	+	−	−
Mare M7 ^1,4,5^ (CVUAS 32689)	B	Died on 10 October 2020	Subcutis of the neck	+	+	+	+
Mare M8 (CVUAS 32656)	A	Euthanized on 24 September 2020	Subcutis of the scapula and abdominal wall	+	−	−	−

^1^ fibrinous peritonitis; ^2^ apostematous pleuritis; ^3^ apostematous myocarditis; ^4^ fibrinous pleuritis; ^5^ apostematous pericarditis; Ln. = lymph node.

**Table 3 animals-12-01612-t003:** Determination of minimal inhibitory concentration (MIC) values (mg/L) for *C. pseudotuberculosis* isolates using the broth microdilution method carried out in microtiter plates. Evaluation of the MIC values was carried out using the MCN6 software (Merlin, Germany). R = resistance, I = intermediary susceptibility, S = susceptibility.

Antibiotic	MIC	Result	Number of Strains	MIC	Result	Number of Strains
Amoxycillin/Clavulanic acid	≤2/1	S	13			
Ampicillin	≤0.25	S	13			
Cefazolin	≤1	S	13			
Cefoxitin	=4	S	11	>4	R	2
Cefquinom	≤2	S	13			
Ceftiofur	=2	S	13			
Cephalothin	≤1	S	13			
Clindamycin	≤0.125	S	13			
Colistin	>2	R	13			
Doxycyclin	≤0.125	S	13			
Enrofloxacin	=0.625	S	13			
Erythromycin	≤0.125	S	13			
Florfenicol	=2	S	13			
Gentamicin	≤1	S	13			
Neomycin	≤8	S	12	>16	R	1
Nitrofurantoin	>64	R	13			
Oxacillin	≥4	R	13			
Penicillin G-potassium	=0.25	I	13			
Rifampin	≤0.0625	S	13			
Spectinomycin	=32	S	9	=64	I	4
Sulfamethoxazol/trimethoprim	≤4.75/0.25	S	13			
Tetracycline	≤0.25	S	13			
Tiamulin	≤0.25	S	13			
Tilmicosin	=2	S	13			
Tulathromycin	≤1	S	13			
Vancomycin	≤0.5	S	13			

**Table 4 animals-12-01612-t004:** Serological testing of alpacas originating from herds A, B and C using a commercial enzyme-linked immunosorbent assay (ELISA) based on a phospholipase D antigen (PLD ELISA) and an in-lab ELISA using a whole cell antigen (WCA ELISA). Serum samples from a total of 232 animals were tested.

	PLD ELISA		WCA ELISA		
Herd	Positive	Negative	Positive	Negative	Total
A	110 (62.9%)	65 (37.1%)	124 (70.9%)	51 (29.1%)	175 (75.4%)
B	8 (20.5%)	31 (79.5%)	8 (20.5%)	31 (79.5%)	39 (16.8%)
C	12 (66.7%)	6 (33.3%)	10 (55.6%)	8 (44.4%)	18 (7.8%)
Total	130 (56%)	102 (44%)	142 (61.2%)	90 (38.8%)	232 (100%)

**Table 5 animals-12-01612-t005:** Comparative serological testing of samples taken from alpacas originating from herds A, B and C using a commercial ELISA based on a phospholipase D (PLD ELISA) and an in-lab ELISA using whole cell antigen (WCA ELISA). A total of 247 serum samples from 232 alpacas were tested.

PLD ELISA	WCA ELISA		Total
	Positive	Negative	
Positive	131 (93.3%)	5 (3.7%)	136 (55.1%)
Negative	21 (18.9%)	90 (81.1)	111 (44.9%)
Total	152 (61.5%)	95 (38.5%)	247 (100%)

## Data Availability

MALDI-TOF MS spectra made in context of this study, were available on exchange basis via the MALDI-User platform MALDI-UP (https://maldi-up.ua-bw.de (accessed on 27 May 2022)).

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
