# Peer review of "Corynebacterium pseudotuberculosis Infections in Alpacas (Vicugna pacos)"

_animals, 2022, doi:10.3390/ani12131612_

Round 1

Reviewer 1 Report

The authors  describe a serological survey of 232 alpacas based on three herds and the post-mortem investigations of four animals derived from this herds.

CPS is regularly found also in alpacas. However there is scarce data on prevalence, clinical outcome, and incubation period in these animals. Moreover, the serological diagnostics used for sheep and goat were proved for their suitability in alpacas.

The manuscipt is well-written. The methods are suitable for the investiagtions. The results are comprehensible presented and adaquately discussed in realtion to the known literature.

There are some points that should be improved/clarified:

1. There is an disagreement withe the number of sera in the abstract. How many sera wered investigated 232 or 247. What is the source of the 15 additional sera. This is not stated in MAterial and Methods.

2. There are two typing errors in the Abstract part, i.e. line 43 pritoneum instead of peritoneum and line 45 seroprvelance instead of seroprevalence.

Author Response

Reviewer 1

  1. There is a disagreement with the number of sera in the abstract. How many sera were investigated 232 or 247. What is the source of the 15 additional sera. This is not stated in Material and Methods.

Author’s comment: 247 sera originated from 232 animals due to multiple testing. This piece of information has been specified in the chapter “Abstract” and “Material and Methods”.

  1. There are two typing errors in the Abstract part, i.e. line 43 pritoneum instead of peritoneum and line 45 seroprvelance instead of seroprevalence.

Author’s comment: We thank the reviewer for pointing out typographical errors. The misspellings have been corrected.

Reviewer 2 Report

General comments:

The manuscript “Corynebacterium pseudotuberculosis Infections in Alpacas (Vi- 2

cugna pacos)” is a well written study (although it has some typos) on the Corynebacterium pseudotuberculosis Infections in alpacas. The manuscript is scientifically sound and the findings are relatively well explained, and it is of interest to the journal. Therefore, I recommend its acceptance for publication after minor reversion.

Specific comments:

1. With regards to the abbreviation for Corynebacterium pseudotuberculosis, “CPS or “C. pseudotuberculosis”? I would use one of these.

2. There are some minor grammatical mistakes and typographical errors in the MS, which are to be corrected.

For example:

Line 122: “animals that had had contact with infected animals were additionally subjected to an”

3. why only 247 alpaca blood samples were collected, does the 247 sample number have a scientific justification? Were samples collected randomly?

4. Meanwhile, 247 samples were collected from 232 animals, you mean some animals were sampled at different time? Such as S8D-S8H? How did you calculate the overall seroprvelance, if S8D-S8H were seropositive for C. pseudotuberculosis? In other word, S8D-S8H were collected from the same animal at different time, if they were all seropositive for C. pseudotuberculosis, then they were deemed as one positive sample or five positive sample )

Author Response

Reviewer 2

Specific comments:

  1. With regards to the abbreviation for Corynebacterium pseudotuberculosis, “CPS” or “C. pseudotuberculosis”? I would use one of these.

Author’s comment: The abbreviation CPS has been replaced by C. pseudotuberculosis.

  1. There are some minor grammatical mistakes and typographical errors in the MS, which are to be corrected.

For example: Line 122: “animals that had had contact with infected animals were additionally subjected to an”

Author’s comment: The manuscript had been reviewed by a native speaker. The wording “had had” was chosen because the animals had contact with infected animals before the animals were tested. For better understanding the word “previously” has been added. The manuscript has been proofread once more by a native speaker.

  1. Why only 247 alpaca blood samples were collected, does the 247 sample number have a scientific justification? Were samples collected randomly?

Author’s comment:  total of 247 sera originating from 232 animals were tested. Of these 15 animals were tested several times.

  1. Meanwhile, 247 samples were collected from 232 animals, you mean some animals were sampled at different time? Such as S8D-S8H? How did you calculate the overall seroprevalence, if S8D-S8H were seropositive for C. pseudotuberculosis? In other word, S8D-S8H were collected from the same animal at different time, if they were all seropositive for C. pseudotuberculosis, then they were deemed as one positive sample or five positive sample).

Author’s comment: Samples from all animals in the three herds older than 6 months were taken. Therefore no calculation of the sample size was performed. This piece information has been included in the chapter “Material and Methods”.

Animals that had been tested at several times were considered positive if one testing yielded a positive result. The seroprevalence was calculated on the basis of the number of animals.

Reviewer 3 Report

The document entitled" Corynebacterium pseudotuberculosis Infections in Alpacas (Vicugna pacos)" is quite novel and does bring important information of diseases in a neo-tropical camelid. There are some changes that must be made to ensure this article can be published.

Specific comments:

Title

The scientific name for alpacas should be properly stated without the hyphen.

Simple summary:

No changes required (good as is)

Abstract:

Line 43 should read 'peritoneum' and not pritoneum.

Introduction: 

Line 77: omit the ')' that closes the citation

Materials and Methods:

No information was taken on sample size calculation. The number of samples taken were stated and the farms. However, there were no sample size calculations to show the reliability of the data, especially in an epidemiological study. This must be included for the information to be published. 

An image of the location of the farms where samples were taken will also add value to the paper for readers.

Results:

Tables should be constant. If "+" and "-" abbreviations are used it should be done throughout. It makes for easier reading to use the symbols rather than the words.

Discussion: 

Line 355, 356 should read 'Histopathological' and not 'pathohistological'

Line 370 should read 'goat' and not 'got'

Line 382: improper in text citation " Schulthess [25]"  and not 'Schulthess (2013) [25]'

Line 403, 406: Braga 2006, 2007 should be reformatted into the journal style. Please omit the year in text. (This should be checked throughout the manuscript). There are too much to place into the report. (especially in the discussion section).

Reference:

This should be revised into the journal format (both in text and in the body)

Author Response

Reviewer 3

Specific comments:

Title

The scientific name for alpacas should be properly stated without the hyphen.

Author’s comment: We had chosen analogous to Braga et al. (2006, 2007) and Sprake et al. (2012) to put the systematic name of alpacas “Vicugna pacos” in brackets.

Simple summary:

No changes required (good as is)

Abstract:

Line 43 should read 'peritoneum' and not pritoneum.

Author’s comment: We thank the reviewer for pointing out this error.

Introduction: 

Line 77: omit the ')' that closes the citation

Author’s comment: The hyphen has been removed.

Materials and Methods:

No information was taken on sample size calculation. The number of samples taken were stated and the farms. However, there were no sample size calculations to show the reliability of the data, especially in an epidemiological study. This must be included for the information to be published.

An image of the location of the farms where samples were taken will also add value to the paper for readers.

Author’s comment: Samples of all animals in the three herds older than 6 months were taken. Therefore no calculation of the sample size was performed. This piece information has been included in the chapter “Material and Methods”.

We promised the animal owners, who participated in this study, that we would publish the data anonymously.

Caseous lymphadenitis in alpacas is a general issue for camelid health throughout Europe, so more detailed data on the location of the participating farms would not be relevant to our opinion.

Results:

Tables should be constant. If "+" and "-" abbreviations are used it should be done throughout. It makes for easier reading to use the symbols rather than the words.

Author’s comment: “positive” has been replaced by “+” and “negative” by “–“ in the tables for better overview.

Discussion:

Line 355, 356 should read 'histopathological' and not “pathohistological”

Author’s comment: “pathohistological” has been replaced by “histopathological”.

Line 370 should read “goats” and not “gots”

Author’s comment: The misspelling has been corrected.

Line 382: improper in text citation "Schulthess [25]" and not “Schulthess (2013) [25]”

Author’s comment: “(2013)” has been removed.

Line 403, 406: Braga 2006, 2007 should be reformatted into the journal style. Please omit the year in text. (This should be checked throughout the manuscript). There are too much to place into the report (especially in the discussion section).

Author’s comment: The years in brackets have been removed.

Reference:

This should be revised into the journal format (both in text and in the body).

Author’s comment: The chapter “References” has been revised.

Round 2

Reviewer 3 Report

The manuscript was revised appropriately and can be published as is.